# Connective tissue disorder and high risk pregnancy: a case series with personalised external aortic root support (PEARS)

Claudia Montanaro [1,2,7] ✉, Polona Kacar[1,7], Giulia Iannaccone[1], John Pepper[2,3], Gurleen Wander[4], Christoph A. Nienaber [2,5], Andreas Hoschtitzky[2,3], Mario Petrou[2,3], Hannah Douglas[6], Mark R. Johnson[2,4], Roshni R. Patel[4], Isma Rafiq[1] & Michael A. Gatzoulis[1,2]

Aortopathy including Marfan (MFS) and Loeys-Dietz syndrome (LDS) poses a high risk of aortic dissection, particularly during pregnancy and the puerperium. Current preventive measures of aortic root dilatation include medical therapy and prophylactic aortic root replacement. The Personalised External Aortic Root Support (PEARS) operation has been developed as an alternative surgical strategy to prevent aortic root dilatation and is now an established procedure with a good prognosis. However, outcomes in pregnant women are unknown. We present case series of nine successful pregnancies in seven women with aortopathy (6 MFS and 1 LDS) who underwent PEARS procedure prior to conception. At a mean follow-up of 4.3 years after delivery, there was no type A or B aortic dissections. Aortic dimensions remained stable, and no hypertensive disorders were observed. Although this is a small retrospective study, PEARS procedure may be a viable pre-conception surgical strategy for women with aortopathy, as an alternative to conventional aortic root surgery. Further studies are needed to conclude that PEARS could be a non-inferior or superior alternative to conventional aortic root surgery in these patients.

Progressive aortic root dilatation and dissection are the leading causes of morbidity and mortality in patients with aortopathy, including Marfan syndrome (MFS) and Loeys–Dietz syndrome (LDS)[1]. The risk of dissection increases with increasing root diameter, therefore preventing aortic root dilatation represents an essential goal. Current preventive measures of aortic root dilatation include medical therapy with beta blockers or angiotensin receptor blockers (ARBs), and prophylactic surgery with aortic root replacement. Based on the most recent guidelines on the aortic disease, young women with a background of MFS and LDS contemplating pregnancy should be considered for aortic root replacement when aortic diameter is ≥45 mm or 40–45 mm if there is evidence of rapid progressive aortic dilatation or a family history of dissection or sudden death[2,3]. Preconception counseling on pregnancy-related risks for such patients is thus paramount[4].

The Personalized External Aortic Root Support (PEARS) operation, first performed in 2004, has been developed as an alternative surgical strategy to prevent aortic root dilatation and minimize the risk of dissection. It entails placing personalized mesh support around the aortic root and the ascending aorta. As of January 2024, 956 patients

[1]Adult Congenital Heart Centre and National Centre for Pulmonary Hypertension, Royal Brompton and Harefield Hospitals, Guy's and St Thomas's NHS Trust, Sydney Street, London, UK. [2]National Heart and Lung Institute, Imperial College, Dovehouse St, London, UK. [3]Cardiothoracic and Congenital Heart Surgery, Royal Brompton and Harefield Hospitals, Guy's and St Thomas's NHS Trust, Sydney Street, London, UK. [4]Department of Obstetrics, Chelsea and Westminster Hospital, 369 Fulham Road, London, UK. [5]Division of Cardiology at the Royal Brompton and Harefield Hospitals, Guy's and St. Thomas's NHS Trust, Sydney Street, London, UK. [6]Department of Cardiology, St Thomas' Hospital, London, UK. [7]These authors contributed equally: Claudia Montanaro, Polona Kacar. ✉ e-mail: cmontanarocardiology@gmail.com

had undergone the PEARS procedure. The principal indication is aortic root dilatation >40 and <50 mm in patients with aortopathy and based on the latest evidence peri-operative mortality is 0.5%, with no reported dissections of the ascending aorta[5]. The PEARS operation has undergone Health Technology Appraisal by the British National Institute for Health and Care Excellence (NICE)[6].

However, the pregnancy outcome of women who underwent the PEARS procedure is unknown, particularly the risk of a type B dissection, which has been reported to be more common in women with Marfan syndrome after aortic root replacement[7].

In this work, we present a detailed case series of nine successful pregnancies after the PEARS procedure performed at the Royal Brompton Hospital, London, UK. At a mean follow-up of 4.3 years after delivery, there were no aortic complications and aortic dimensions remained stable.

## Results

Seven women post PEARS had nine successful pregnancies at the mean age of 33 ± 5.3 years; 6 had a genetic diagnosis of MFS and one of LDS type 4 (Fig. 1). Indication for PEARS was aortic root dilatation ≥45 mm in 4 patients, whereas in the three remaining PEARS was performed in anticipation of pregnancy (mean aortic root dimension 41.7 ± 1.5 mm). Mean age at the time of PEARS was 26.8 ± 6.9 years with the median interval between PEARS and confirmed pregnancy 2.6 (1.2–15.6) years.

At a mean follow-up of 4.3 years (range: 2 months–14.8 years) after delivery, there were no type A or B aortic dissections, or maternal death, or the need for aortic reintervention. The aortic root, aortic

arch, and descending aorta dimensions were monitored during and after pregnancy with echocardiography or magnetic resonance imaging, with stable measurements (Table 1). Blood pressure (BP) remained well controlled (mean BP 108 ± 13.6/66 ± 12.1); no-one developed pre-eclampsia. Beta-blockers were used in 6 pregnancies (67%) and were well tolerated.

Six deliveries were by Cesarean section (CS) (five elective, one emergency at 31 + 2 weeks following spontaneous onset of labor). The remaining three vaginal deliveries were performed under epidural analgesia, one requiring forceps assistance.

## Discussion

This case series of nine successful pregnancies in seven women with underlying aortopathy who underwent the PEARS procedure prior to conception demonstrates no aortic complications during pregnancy and at a mean follow-up of 4.3 years post-delivery for this high-risk patient cohort. Aortic dimensions were assessed periodically and remained stable in the antenatal and postpartum periods in all women. No hypertensive disorders of pregnancy were reported. No type A or B aortic dissection was noted at the last follow-up.

Mitigating the increased risk of aortic dissection in MFS and LDS is the major objective in the care of these high-risk patients. Medical therapy, in particular beta blockers and ARBs, is recommended according to the recent guidelines[3,8], but ARBs are not recommended during pregnancy. Elective aortic root surgery may be considered and should be discussed with these patients when contemplating pregnancy. Current surgical options include Bentall operation where both

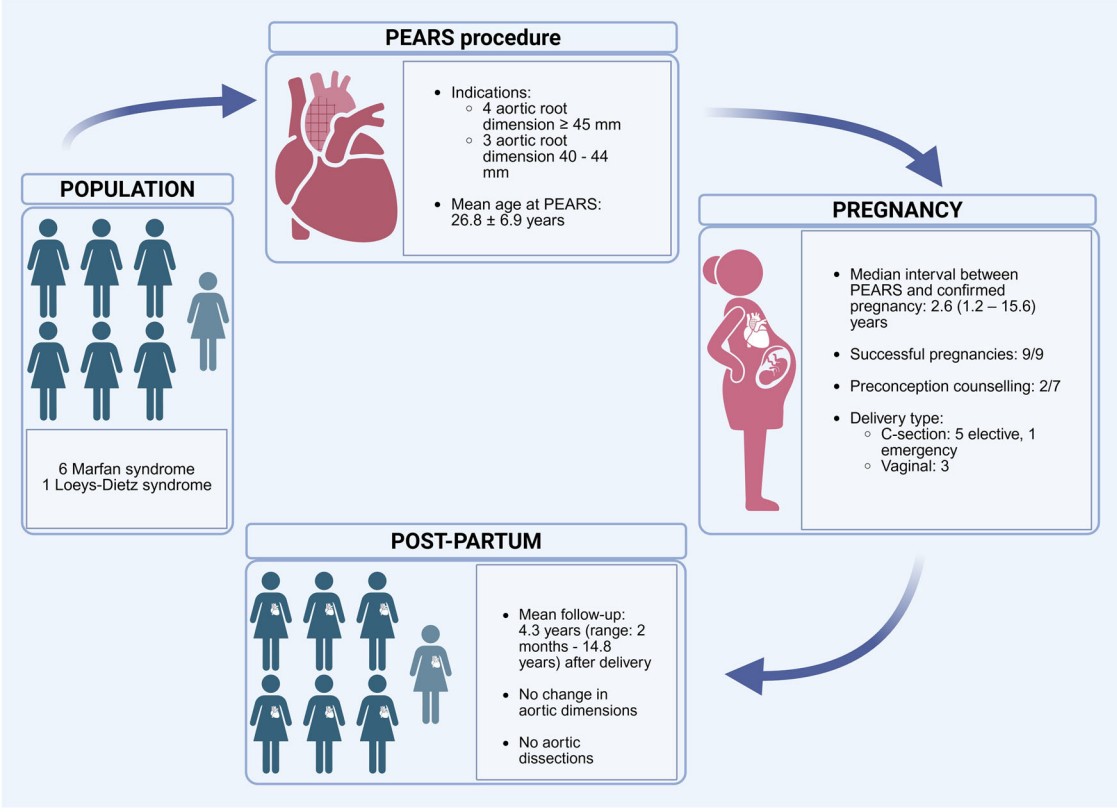

**Fig. 1 | Pregnancy outcomes following PEARS in women with heritable thoracic aortic disease.** Schematic overview of pregnancy outcomes in women with Marfan syndrome (*n* = 6) and Loeys–Dietz syndrome (*n* = 1) following the PEARS procedure. Indication for PEARS was aortic root dilatation ≥45 mm (4 patients) and anticipation of pregnancy (3 patients). Mean age at the time of PEARS was 26.8 ± 6.9 years with the median interval between PEARS and confirmed pregnancy 2.6

(1.2–15.6) years. All 9 pregnancies were successful. Delivery methods included 6 Cesarean sections (5 elective, 1 emergency) and 3 vaginal deliveries. During a mean follow-up of 4.3 years (range: 2 months–14.8 years) after delivery, there were no type A or B aortic dissections, and aortic dimensions remained stable. PEARS Personalized External Aortic Root Support. Created in BioRender. Kacar, M. (2025) https://BioRender.com/k9f5hko.

**Table 1 | Pregnancy outcomes in women with Marfan syndrome or Loeys–Dietz syndrome following PEARS procedure**

| Patient (pregnancy number) | Underlying Diagnosis | Indication for PEARS | Age at PEARS | Ethnicity | BMI | Chronic hypertension | Delivery type | Beta-blockers during pregnancy | Aortic Dissection related to pregnancy† | Aortic dimensions (mm) | | | | | |
| --- | --- | --- | --- | --- | --- | --- | --- | --- | --- | --- | --- | --- | --- | --- | --- |
| | | | | | | | | | | Pre-pregnancy | | | Post-pregnancy | | |
| | | | | | | | | | | AoR[a] | Arch[b] | DescAo[c] | AoR[a] | Arch[b] | DescAo[c] |
| 1 (1) | LDS | AoR 40 mm, future pregnancy | 32 | Asian | 32.3 | No | Elective C-section | Yes | No | 39 | 29 | 17 | 39 | 30 | 18 |
| 1 (2) | | | | | | | Elective C-section | Yes | No | 41 | 30 | 18 | 40 | 27 | 17 |
| 2 | MFS | AoR 45 mm | 22 | White | 33.7 | No | Emergency C-section* | Yes | No | 44 | 28 | 20 | 44 | 27 | 18 |
| 3 | MFS | AoR 48 mm | 20 | White | 23.1 | No | Elective C-section | No | No | 46 | 35 | 18 | 46 | 35 | 20 |
| 4 | MFS | AoR 45 mm | 26 | White | 16.6 | No | Elective C-section | Yes | No | 43 | 20 | 16 | 41 | 21 | 16 |
| 5 (1) | MFS | AoR 47 mm | 20 | White | 23.5 | No | Vaginal assisted | No | No | 47 | 33 | 18 | 47 | 35 | 19 |
| 5 (2) | | | | | | | Elective C-section | Yes | No | 47 | 35 | 19 | 47 | 35 | 19 |
| 6 | MFS | AoR 43 mm, History of AoR dilatation during previous pregnancy | 35 | White | 20.7 | No | Vaginal | Yes | No | 37 | 33 | 18 | 37 | 30 | 22 |
| 7 | MFS | AoR 42 mm, future pregnancy | 32 | White | 19.7 | No | Vaginal | No | No | 45 | 26 | 18 | 45 | 26 | 18 |

LDS Loeys–Dietz syndrome, MFS Marfan syndrome, PEARS Personalized External Aortic Root Support, BMI body mass index, AoR aortic root, DescAo descending aorta.

†Aortic dissection occurring during pregnancy or up to 6 weeks post-partum; [a]p = 0.93; [b]p = 0.96; [c]p = 0.52; the Wilcoxon signed-rank test was used to compare continuous variables. *Spontaneous rupture of membranes and premature labor.

aortic valve and aortic root are replaced with composite valve conduit, and valve-sparing root replacement surgery (VSRR). If a mechanical valve prosthesis is used during Bentall operation, lifelong anticoagulation is required, which may further increase maternal and fetal complications. The VSRR operation does not completely eliminate the risk of cardiac complications during pregnancy as peripartum type B dissection has been reported[7,9].

Data pertaining effectiveness of the PEARS procedure during pregnancy is scarce. There are only two individual case reports describing uncomplicated delivery via CS after PEARS[10]. In one, the PEARS operation was performed during the second trimester[11]. Outside of pregnancy, research indicates that PEARS is a viable pre-emptive intervention in patients with genetic aortopathies and moderate aortic root dilation as it has the potential to stabilize the aortic root and to prevent its further dilatation. It stabilizes the aortic root while becoming incorporated histologically, thus further reducing wall stress[12]. According to the recent most extensive report on PEARS, only one asymptomatic type B dissection was reported 3 years after PEARS procedure[5].

Our case series of pregnancy after the PEARS procedure introduces an alternative pre-conception surgical strategy for women with aortopathy, especially considering the low procedural mortality risk. For women not meeting the criteria for prophylactic surgery, medical therapy represents the only recommended preventive measure despite the lack of strong evidence[13]. Of note, 3 women in our cohort underwent the PEARS procedure in anticipation of pregnancy, albeit not reaching the classic criteria for pre-emptive surgery. In our cohort, the pregnancy was not associated with aortic complications either with vaginal delivery or CS.

Pre-pregnancy counseling in women with aortopathy should include discussing the PEARS procedure if aortic surgery is deemed appropriate. In our study, the PEARS procedure was not associated with aortic complications during pregnancy or in the puerperium. However, this is a retrospective study with a small sample size and longer, prospective, multicenter studies are necessary to further evaluate the safety and efficacy of the PEARS procedure in this population.

## Methods

A retrospective cohort study included women with successfully completed pregnancies after the PEARS procedure. Demographic, imaging (echocardiography, magnetic resonance imaging), medication, and delivery modality data were collected before and after pregnancy as appropriate.

The study was performed according to the Declaration of Helsinki. Data was collected for routine clinical practice and administrative purposes, and therefore, the ethics committee waived the need for approval for this retrospective non-interventional study using the NHS-wide Health Research Authority software (www.hra-decisiontools.org.uk/research/). PEARS is a routine procedure offered at our institution and was offered to the women included in this case series as part of their clinical care. The authors affirm that the patients provided written informed consent for the publication of the medical information included in this paper.

The datasets generated and/or analyzed during the current study include sensitive patient information and are subject to ethical and privacy restrictions. As such, they are not publicly available but can be shared by the corresponding author upon justified request and subject to institutional and ethical approvals.

This study included only participants who were biologically female. The sex of participants was determined based on self-report at the time of recruitment. Gender identity was not separately assessed, as the clinical relevance of the study was tied specifically to biological sex in the context of post-pregnancy cardiovascular health and surgical outcomes.

As all participants identified as women, no sex- or gender-based comparative analysis was conducted. Data have not been disaggregated by sex or gender, as the study cohort was homogenous in this regard.

## Statistical analysis

Continuous variables are presented as mean ± SD or median (range), and categorical variables as numbers and percentages. The Wilcoxon signed-rank test was used to compare continuous variables. All statistical analyses were performed using SPSS version 27.0 software. Values of $p < 0.05$ were considered statistically significant.

## Reporting summary

Further information on research design is available in the Nature Portfolio Reporting Summary linked to this article.

## Data availability

The datasets generated and/or analyzed during the current study include sensitive patient information and are subject to ethical and privacy restrictions. As such, they are not publicly available but may be shared by the corresponding author upon reasonable request for purposes such as research validation or verification, and contingent on approval from the relevant institutional review board or ethics committee. Please allow up to 4 weeks for a response to data access requests.

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

## Author contributions

C.M. and P.K. contributed to the study's conception and design. Material preparation, data collection, and analysis were performed by I.G., R.R.P., H.D., and G.W. The first draft of the manuscript was written by C.M. and P.K. J.P., C.A.N., A.H., M.P., M.R.J., I.R., and M.A.G. critically revised and commented on all versions of the manuscript. All authors read and approved the final manuscript.

## Competing interests

The authors declare no competing interests.
