## [Peer Review File · Nature Communications]

Connective tissue disorder and high risk pregnancy: a case series with personalised external aortic root support (PEARS)

Corresponding Author: Dr Claudia Montanaro

Version 1:

Reviewer comments:

Reviewer #1

(Remarks to the Author)

This is a case series of 7 individuals (6MFS, 1 LDS) had 9 successful pregnancies after PEARS. Indication was aortic root dilation >45 mm in 4 persons and 3 had mean aortic root dimensions of 41.7+1.5 mm. Mean interval between PEARS and delivery 5 years (1.5-8 yrs). Mean f/u after delivery was 4.7 yrs with no type A and no type B dissections, no maternal deaths nor need for aortic intervention. This is still a small study but does add information to the literature about a potential alternative treatment for prevention of aortic dissection in pregnancy. Further, larger studies need to be done to prove this is a safe alternative.

Questions/Comments:

1. Can the authors (in line 110) give not only the mean follow-up for patients after delivery but also range?
2. With PEARS, do we expect to see aortic growth with wrap in place in a short interval? Lines 124-125 and Table 1? Looking at root measurements they mostly stayed the same or decreased, there was small changes in arch and descending measurements- statistically no change in measurements, but is this expected with wrap in place? Please put in the table units (mm) I assume this was mm but cannot see unit measurements for aortic dimensions. Additionally it is important to further detail "pre-pregnancy" vs "post-pregnancy" measurements; was there standardized timing of when these measurements were done pre and post pregnancy?
3. Table 1- what are authors defining as aortic dissection related to pregnancy- dissection within 12 weeks? 6 months? 1 year? This needs to be defined
4. Can the authors add neonatal data in terms of GA at delivery, birth weights and percentiles as safety in pregnancy also implies that this is safe for neonates and having some fetal/neonatal data would be helpful as well. But again larger studies need to be done.
5. It is important to clarify the timing from PEARS to conception/early pregnancy as this is the more important timing interval of PEARS and then a future pregnancy in lines 107-109, not necessarily PEARS to delivery. The PEARS to conception is more important as this information is clinically relevant for counseling related to attempting pregnancy after PEARS procedure. Please provide if possible interval from PEARS to conception or confirmed pregnancy.
6. Finally, the risk for type A and B dissection is only 2-3% in modern cohorts with MFS/LDS with these aortic sizes and thus it is hard to conclude anything about safety or risks of dissection with such a small cohort. It is important in the conclusions that authors do not overstate results and state we cannot use this information to say that it prevents type A and B dissection but none have been seen in this small cohort.

Reviewer #2

(Remarks to the Author)

General Comments: The authors describe 9 pregnancies in 7 women who had previously undergone the PEARS procedure, 3 of 7 in anticipation of pregnancy. During almost 5 years of follow-up, no aortic dissections occurred. The literature on pregnancy following root surgery in genetic aortopathies is quite sparse, so this report is of value. One obvious question is whether the PEARS procedure is superior to standard root replacement, particularly as PEARS is not yet widely available (and not yet approved in the US). The authors do not speculate about this question but only state that root replacement was associated with a 3.4% risk of type B dissection in a UK study. My reading of the Cauldwell study indicates that of 5

pregnancy-related dissections, one occurred in a woman with previous root replacement (of 14 with previous root replacement). With such small numbers, percentages may be deceptive. The authors do not cite an earlier study (AJP Rep 2019) of 14 Marfan women with 20 pregnancies and previous root replacement in whom no dissections occurred. Nevertheless, the apparent low rate of type B dissection following PEARS in general compared to standard root replacement is intriguing and suggests more favorable post-procedure flow dynamics.

Specific Comments:

1. Reference 5 is missing the journal title.
2. The authors may wish to add more information about pregnancy outcomes, e.g., were all births singletons, were any affected, was IVF and pre-implantation testing done before any of the pregnancies, was use of beta blockade (in 6 of 9 pregnancies) associated with any fetal growth retardation?

Reviewer #3

(Remarks to the Author)

Connective tissue disorder and high-risk pregnancy: outcomes with personalised external aortic root support (PEARS)

Aortic dissection is a leading cause of maternal death. Women with genetic aortopathies are at high risk of aortic dissection during and after pregnancy, especially if their aorta is already dilated. This single centre retrospective case series by Montanaro et al describes seven women with genetic aortopathy syndromes who underwent personalised external aortic root support (PEARS) procedures prior to and in anticipation of pregnancy. Between them, they went on to have nine successful pregnancies subsequently. There were no aortic dissections during or after pregnancy and during follow-up (mean 4.7 years). Of the seven, three women with dilated aortas were operated before reaching guideline diameters for intervention. Conventionally, patients are treated by preemptive aortic root replacement at threshold size, however in the expectation that the haemodynamic stress of pregnancy may cause accelerated disease progression early intervention may be justified in this high-risk cohort. There is currently not enough evidence to confirm that this improves pregnancy outcomes in such patients because the numbers are small.

The evidence base supporting this procedure world-wide remains small, although medium-term outcomes and complication rates are favourable.(1) There are only two case reports relating to PEARS and pregnancy, Frost et al 2020 (2) and Kenny et al (3).

This case series therefore contributes to a small, but growing amount of evidence supporting PEARS. In the context of pregnancy, this novel approach to pre-conception optimisation could be offered to women with high-risk aortopathy syndromes in whom pregnancy would usually be counselled against.(4) More long-term outcome data are needed to advocate this technique alongside or even instead of conventional root replacements, in line with current guidelines and as the authors themselves conclude.

The data are presented clearly, with appropriately limited use of statistics (n=7). The references cited are up to date and relevant.

References:

- (1) Van Hoof, L., Rega, F., Golesworthy, T., Verbrugghe, P., Austin, C., Takkenberg, J.J., Pepper, J.R. and Treasure, T., 2021. Personalised external aortic root support for elective treatment of aortic root dilation in 200 patients. *Heart*, 107(22), pp.1790-1795.
- (2) Frost, C., Williams, L., Naidoo, R., Smith, I. and Tesar, P., 2020. 617 PEARS: an Alternate Procedure in Pregnant Patients With Marfan Syndrome. *Heart, Lung and Circulation*, 29, p.S315.
- (3) Kenny, L.A., Austin, C., Golesworthy, T., Venugopal, P. and Alphonso, N., 2021. Personalized external aortic root support (PEARS) for aortic root aneurysm. *Operative Techniques in Thoracic and Cardiovascular Surgery*, 26(2), pp.290-305.
- (4) Regitz-Zagrosek, V., Roos-Hesselink, J.W., Bauersachs, J., Blomström-Lundqvist, C., Cifkova, R., De Bonis, M., Jung, B., Johnson, M.R., Kintscher, U., Kranke, P. and Lang, I.M., 2018. 2018 ESC guidelines for the management of cardiovascular diseases during pregnancy: the task force for the management of cardiovascular diseases during pregnancy of the European Society of Cardiology (ESC). *European heart journal*, 39(34), pp.3165-3241.

Version 2:

Reviewer comments:

Reviewer #2

(Remarks to the Author)

Reviewer #3

(Remarks to the Author)

Most of the reviewers' points have been satisfactorily addressed.

The requests pertaining to detailed obstetric and neonatal data have not been addressed because the data are unavailable as patients delivered in local obstetric units.

To:

Riikka Jokinen, PhD

Senior Editor

Rome, 9th February 2025

Dear Editor and Reviewers,

We thank the Editor and the Reviewers for evaluating our manuscript and for their constructive feedback. We submit herewith our response addressing the points raised by the reviewers. Please, find the answers below and the corrections highlighted in the text of the revised manuscript.

Reviewer 1:

This is a case series of 7 individuals (6MFS, 1 LDS) had 9 successful pregnancies after PEARS. Indication was aortic root dilation >45 mm in 4 persons and 3 had mean aortic root dimensions of 41.7+1.5 mm. Mean interval between PEARS and delivery 5 years (1.5-8 yrs). Mean f/u after delivery was 4.7 yrs with no type A and no type B dissections, no maternal deaths nor need for aortic intervention. This is still a small study but does add information to the literature about a potential alternative treatment for prevention of aortic dissection in pregnancy. Further, larger studies need to be done to prove this is a safe alternative.

Questions/Comments:

1. Can the authors (in line 110) give not only the mean follow-up for patients after delivery but also range?

- The range was added to the mean follow-up for patients after delivery. Please see the corresponding change within the text:

At a mean follow-up of 4.3 years (range: 2 months – 14.8 years) after delivery, there was no type A or B aortic dissections, nor maternal death or the need for aortic reintervention.

2. With PEARS, do we expect to see aortic growth with wrap in place in a short interval?

Lines 124-125 and Table 1? Looking at root measurements they mostly stayed the same or decreased, there was small changes in arch and descending measurements- statistically no change in measurements, but is this expected with wrap in place? Please put in the table units (mm) I assume this was mm but cannot see unit measurements for aortic dimensions. Additionally, it is important to further detail “pre-pregnancy” vs “post-pregnancy” measurements; was there standardized timing of when these measurements were done pre and post pregnancy?

- Thank you for this valuable comment. We agree with you that the wrapped aortic segment is unlikely to change during time but we measured it for completeness and because this is the first case series. Indeed, the concern is more about the segment post PEARS, for two reasons: 1. These patients have connective tissue disease which can manifest during pregnancy, 2. One could speculate that the presence of a rigid wrap increases the chance of distal dissection.
- The units were added in the table.
- The last imaging before pregnancy was considered as pre-pregnancy and the first imaging after delivery was considered as post-pregnancy; however, there was no standardized timing of when these examinations were performed.

3. Table 1- what are authors defining as aortic dissection related to pregnancy- dissection within 12 weeks? 6 months? 1 year? This needs to be defined.

- Thank you for outlining this. Aortic dissection related to pregnancy was defined as any dissection occurring during pregnancy or up to 6 weeks post-partum. Please see the corresponding definition added to the Table 1.

4. Can the authors add neonatal data in terms of GA at delivery, birth weights and percentiles as safety in pregnancy also implies that this is safe for neonates and having some fetal/neonatal data would be helpful as well. But again larger studies need to be done.

- Another very valuable comment. We agree that foetal/neonatal data are of great importance when discussing safety during pregnancy, and this was also discussed at length among the authors. However, due to patients delivering in local hospitals outside London, we were unable to obtain verifiable obstetric data from many women included in this study. Therefore, we omitted these data and focused on cardiovascular outcomes instead.

5. It is important to clarify the timing from PEARS to conception/early pregnancy as this is the more important timing interval of PEARS and then a future pregnancy in lines 107-109, not necessarily PEARS to delivery. The PEARS to conception is more important as this information is clinically relevant for counseling related to attempting pregnancy after PEARS procedure. Please provide if possible interval from PEARS to conception or confirmed pregnancy.

- The interval from PEARS to confirmed pregnancy was added in the text:

Mean age at the time of PEARS was 26.8 ± 6.9 years with the median interval between PEARS and confirmed pregnancy 2.6 (1.7 – 15.6) years.

6. Finally, the risk for type A and B dissection is only 2-3% in modern cohorts with MFS/LDS with these aortic sizes and thus it is hard to conclude anything about safety or risks of dissection with such a small cohort. It is important in the conclusions that authors do not overstate results and state we cannot use this information to say that it prevents type A and B dissection but none have been seen in this small cohort.

- Although this is a small cohort, to our knowledge it is the largest cohort of pregnant women after PEARS procedure reported in the literature so far. We agree that no firm conclusions can be drawn given the small sample size. To further emphasize this, we added the following sentence in the conclusions:

Although the PEARS procedure was not associated with aortic complications during pregnancy or in the puerperium in our cohort, the sample size is small, and longer, prospective, multicenter studies are necessary to further evaluate the safety and efficacy of the PEARS procedure in this population.

Reviewer #2:

General Comments: The authors describe 9 pregnancies in 7 women who had previously undergone the PEARS procedure, 3 of 7 in anticipation of pregnancy. During almost 5 years of follow-up, no aortic dissections occurred. The literature on pregnancy following root surgery in genetic aortopathies is quite sparse, so this report is of value. One obvious question is whether the PEARS procedure is superior to standard root replacement, particularly as PEARS is not yet widely available (and not yet approved in the US). The authors do not speculate about this question but only state that root replacement was associated with a 3.4% risk of type B dissection in a UK study. My reading of the Cauldwell study indicates that of 5

pregnancy-related dissections, one occurred in a woman with previous root replacement (of 14 with previous root replacement). With such small numbers, percentages may be deceptive. The authors do not cite an earlier study (AJP Rep 2019) of 14 Marfan women with 20 pregnancies and previous root replacement in whom no dissections occurred. Nevertheless, the apparent low rate of type B dissection following PEARS in general compared to standard root replacement is intriguing and suggests more favorable post-procedure flow dynamics.

- Thank you for this valuable comment. We agree that percentages may be deceptive in this case; however, we wanted to emphasize that prophylactic root replacement does not negate cardiac complications in these patients. Nevertheless, to prevent any misunderstanding, we adapted the text accordingly:

The VSRR operation does not completely eliminate the risk of cardiac complications during pregnancy as peripartum type B dissection has been reported^{7,9}.

Specific Comments:

1. Reference 5 is missing the journal title.
 - Thank you for pointing this out. The reference was corrected.
2. The authors may wish to add more information about pregnancy outcomes, e.g., were all births singletons, were any affected, was IVF and pre-implantation testing done before any of the pregnancies, was use of beta blockade (in 6 of 9 pregnancies) associated with any fetal growth retardation?
 - Thank you for this comment. As many patients delivered in local hospitals outside London, we were unable to obtain verifiable obstetric data from a significant number of women included in this study. Therefore, we omitted these data and instead focused on cardiovascular outcomes.

Reviewer #3:

Connective tissue disorder and high-risk pregnancy: outcomes with personalised external aortic root support (PEARS)

Aortic dissection is a leading cause of maternal death. Women with genetic aortopathies are at high risk of aortic dissection during and after pregnancy, especially if their aorta is already

dilated. This single centre retrospective case series by Montanaro et al describes seven women with genetic aortopathy syndromes who underwent personalised external aortic root support (PEARS) procedures prior to and in anticipation of pregnancy. Between them, they went on to have nine successful pregnancies subsequently. There were no aortic dissections during or after pregnancy and during follow-up (mean 4.7 years). Of the seven, three women with dilated aortas were operated before reaching guideline diameters for intervention. Conventionally, patients are treated by preemptive aortic root replacement at threshold size, however in the expectation that the haemodynamic stress of pregnancy may cause accelerated disease progression early intervention may be justified in this high-risk cohort. There is currently not enough evidence to confirm that this improves pregnancy outcomes in such patients because the numbers are small.

The evidence base supporting this procedure world-wide remains small, although medium-term outcomes and complication rates are favourable.⁽¹⁾ There are only two case reports relating to PEARS and pregnancy, Frost et al 2020 ⁽²⁾ and Kenny et al ⁽³⁾. This case series therefore contributes to a small, but growing amount of evidence supporting PEARS. In the context of pregnancy, this novel approach to pre-conception optimisation could be offered to women with high-risk aortopathy syndromes in whom pregnancy would usually be counselled against.⁽⁴⁾ More long-term outcome data are needed to advocate this technique alongside or even instead of conventional root replacements, in line with current guidelines and as the authors themselves conclude. The data are presented clearly, with appropriately limited use of statistics (n=7). The references cited are up to date and relevant.

References:

- (1) Van Hoof, L., Rega, F., Golesworthy, T., Verbrugghe, P., Austin, C., Takkenberg, J.J., Pepper, J.R. and Treasure, T., 2021. Personalised external aortic root support for elective treatment of aortic root dilation in 200 patients. *Heart*, 107(22), pp.1790-1795.
- (2) Frost, C., Williams, L., Naidoo, R., Smith, I. and Tesar, P., 2020. 617 PEARS: an Alternate Procedure in Pregnant Patients With Marfan Syndrome. *Heart, Lung and Circulation*, 29, p.S315.
- (3) Kenny, L.A., Austin, C., Golesworthy, T., Venugopal, P. and Alphonso, N., 2021. Personalized external aortic root support (PEARS) for aortic root aneurysm. *Operative Techniques in Thoracic and Cardiovascular Surgery*, 26(2), pp.290-305.
- (4) Regitz-Zagrosek, V., Roos-Hesselink, J.W., Bauersachs, J., Blomström-Lundqvist, C., Cifkova, R., De Bonis, M., Iung, B., Johnson, M.R., Kintscher, U., Kranke, P. and Lang, I.M.,

2018. 2018 ESC guidelines for the management of cardiovascular diseases during pregnancy: the task force for the management of cardiovascular diseases during pregnancy of the European Society of Cardiology (ESC). *European heart journal*, 39(34), pp.3165-3241.